# Dissatisfaction with Local Medical Services for Middle-Aged and Elderly in China: What Is Relevant?

**DOI:** 10.3390/ijerph18083931

**Published:** 2021-04-08

**Authors:** Xiaojing Fan, Min Su, Yaxin Zhao, Duolao Wang

**Affiliations:** 1School of Public Policy and Administration, Xi’an Jiaotong University, Xi’an 710049, China; fanxj112@xjtu.edu.cn; 2School of Public Administration, Inner Mongolia University, Hohhot 010021, China; 3School of Public Health, Xi’an Jiaotong University Health Science Center, Xi’an 710061, China; zhaoyaxin1996@stu.xjtu.edu.cn; 4Department of Clinical Sciences, Liverpool School of Tropical Medicine, Liverpool L3 5QA, UK; duolao.wang@lstmed.ac.uk

**Keywords:** dissatisfaction, local medical services, determinants, binary outcome, middle-aged and elderly person

## Abstract

As violent clashes between doctors and patients in China intensify, patient dissatisfaction has been identified as a major concern in the current healthcare reform in China. This study aims to investigate the main determinants of dissatisfaction with local medical services attributable to middle-aged and elderly characteristics and identify areas for improvement. A total of 14,263 rural participants and 4898 urban participants were drawn from the China Health and Retirement Longitudinal Study in 2018. Dissatisfaction was measured by two methods: binary outcome (1 = Dissatisfaction; 0 = No) demonstrated the risk of occurring dissatisfaction among various characteristics, and continuous outcome (ranges from score 1 to 5) showed the degree. The mean score of dissatisfaction was 2.73 ± 1.08. Sixteen percent of rural participants and 19% of urban participants reported dissatisfaction with local medical services, respectively. The multilevel analyses demonstrated that participants’ utilization of paid family doctor services decreased the risk of occurring dissatisfaction; dissatisfaction was less focused on females; having chronic diseases increased the risk of dissatisfaction. This study suggests promotion of family doctor services can effectively reduce middle-aged and elderly dissatisfaction with the local medical services. In addition, more attention should be focused on males and middle-aged and elderly with chronic diseases in order to decrease dissatisfaction.

## 1. Introduction

The views of patients are becoming increasingly important in the process of improving the health system [1,2]. Patient dissatisfaction refers to the attitude or feeling of being unsatisfied, displeased or disappointed when using healthcare services [3]. It is a measure of the level of content of care they receive from medical institutions and is a significant performance dimension that provides healthcare managers and professionals with useful insights for improving the quality and effectiveness of care [2,4,5,6]. In China, despite a range of health policies to provide greater health benefits to citizens, patient dissatisfaction with medical services continues to rise and can even escalate into serious incidents of violence against doctors. Studies show that violence in healthcare facilities has increased 10 times in the last 10 years [7]. The growing tension between patients and healthcare providers has challenged China’s current healthcare reforms, making patient dissatisfaction one of the primary concerns of the health system.

Based on a literature search, we found that most dissatisfaction surveys are based on quantitative cross-sectional studies [2,5,8,9], cohort studies [10] or qualitative interviews [11]. Several studies have investigated that patient satisfaction is shaped by some determinants including personal characteristics, preference, expectation and the quality of care received in other countries based on cross-sectional surveys [5,12,13,14]. In addition, a cross-sectional study in Switzerland found that the most satisfied patients were the most likely to participate in a post-hospitalization satisfaction survey [8]. In Pakistan, a low level of patient satisfaction was found to be correlated with the medical services [15]; scholars found that the development of the interpersonal and clinical skills of doctors could improve patient satisfaction [16]. A cross-sectional study in Italian university hospitals also found that lower patient satisfaction was associated with a higher patients leaving hospital against medical advice rates [2]. Studies that emerged from China recently are only limited to selected groups, tertiary public hospitals, certain types of treatments or health conditions [17]. For example, in a sample of 300 inpatients in one tertiary public hospital it was found that the discharge and admission process does not yet meet patient expectations [18]. To investigate provider-related factors, a cross-sectional study found that higher levels of trust, lower levels of hospital medical expenditure and good staff attitude were key predictors of patient satisfaction [9]. Some scholars studied the impact of both patient- and provider-related characteristics on patient satisfaction in urban China by a cohort study [10].

The previously mentioned studies provide useful insight. However, there is limited evidence from both urban and rural China focusing on the patients’ dissatisfaction and to what degree patient characteristics explain dissatisfaction among middle-aged and elderly. With the rapid development of the Chinese economy, people aged 60 years and over accounted for 13.26% of the total, according to the result of the sixth national census, up 2.93 percentage points from the fifth national census, and 8.87% of people aged 65 and over, up 1.91 percentage points from the fifth national census [19]; the aging process is beginning to accelerate gradually. Studies have found that aging largely alters the functional capacity of the human body [20,21] and is strongly associated with the incidence of common diseases, chronic illnesses and cancer [22,23,24]. With the changing age structure of the world’s population and the gradual increase in aging in China, the middle-aged and elderly population will make up the majority of the groups utilizing healthcare services in the future, research on dissatisfaction among them would be an important reference for targeted efforts to improve the health care experience of the target population, promote a good health care environment and reduce social risks. All of these arguments mentioned above make our study an innovative and original piece of work.

In this study, we explored patients’ dissatisfaction with local health services among middle-aged and elderly people by adopting a nationally representative cross-sectional survey data in China. The findings uncover the level of patients’ dissatisfaction from the largest and most populated developing countries, investigate the relationship between patients’ dissatisfactions and their characteristics (such as sociodemographic characteristics and health services utilization) in rural and urban areas and reveal the relative significant reasons for dissatisfaction. The ultimate aim is to help managers in hospitals and policy-makers to take target measures for improving patient satisfaction. In the following sections, the sources of the research data, the selection of key variables and the methods of analysis, the findings and the discussion are presented separately.

## 2. Methods

### 2.1. Data Sources

Data were obtained from nationally representative face-to-face household surveys in 2018, the China Health and Retirement Longitudinal Study (CHARLS). The conceptual basis and design of the CHARLS have been described extensively in the literature [25,26]. The CHARLS utilizes a multi-stage probability proportional scale sampling method to randomly select Chinese middle-aged and elderly and their spouses from 150 counties and 450 communities/villages across 28 provinces as respondents [27]. In this study, a total of 14,263 rural participants and 4898 urban participants aged more than 45 years old were selected. Based on previous theoretical studies on the process of patient satisfaction formation [10,28,29,30], research on the determinants affecting patient dissatisfaction can help improve patient evaluations and therefore patient satisfaction. The framework for the empirical exploration of patient dissatisfaction in this paper is shown in Figure 1.

### 2.2. Variables

Individual dissatisfaction was the dependent variable, and it was based on the middle-aged and elderly reports in the 2018 wave on the following item: “Are you satisfied with the quality, cost and convenience of local medical services? Choose from very satisfied (1), somewhat satisfied (2), neutral (3), somewhat dissatisfied (4) and very dissatisfied (5)”. Two methods were used to analyze dissatisfaction. Firstly, we grouped dissatisfaction as a binary variable, whereby 0 denoted “no”, including “very satisfied, somewhat satisfied and neutral”; 1 denoted “yes” including “somewhat dissatisfied and very dissatisfied”. Secondly, we treated dissatisfaction as a continuous variable, and it was measured by the above item on a five-point Likert scale and ranged from score 1 to 5. We assigned a score to each category: score 1 for “very satisfied”, score 2 for “somewhat satisfied”, score 3 for “neutral”, score 4 for “somewhat dissatisfied” and score 5 for “very dissatisfied”. The higher the score, the higher the level of dissatisfaction. The independent variables were sociodemographic factors (participants’ sex, age, education, economic and living status and chronic diseases) and utilization of outpatient, inpatient and paid family doctor services based on previous studies but constrained by the variables collected in the CHARLS (Table 1) [10,13,31,32].

### 2.3. Data Analysis

Multilevel mixed-effect models were applied to explore the determinants for dissatisfaction with local medical services from the individual’s perspective. These determinants in Table 1 were specified as the fixed effect, and the community where participants lived was a random effect. The odds ratios (ORs) with 95% confidence limits (CLs) in multilevel mixed-effects logistic regression (model 1) demonstrated the risk of occurring dissatisfaction among various characteristics, and the coefficients in multilevel mixed-effects linear regression with 95% CLs (model 2) showed the extent to which participants’ dissatisfaction was associated with various characteristics. The statistical results and figures were processed with STATA statistical software version 14.0 (StataCorp LP, College Station, TX, USA) and Excel 2016, respectively. A two-tailed *p*-value of <0.05 was considered statistically significant.

## 3. Results

### 3.1. Descriptive Statistics

Figure 2 indicates that a total of 9% and 7% of participants reported “very dissatisfied” and “somewhat dissatisfied” with local medical services, respectively, while 46% of participants had a “neutral (neither satisfied nor dissatisfied)” attitude toward local medical services. After aggregating groups based on “somewhat dissatisfied” and “very dissatisfied” replies, overall 16% of rural participants and 19% of urban participants reported dissatisfaction with local medical services, respectively, while 44% of rural participants and 50% of urban participants had a “neutral” attitude toward local medical services, respectively.

Table 2 presents the summary statistics of basic variables in our sample. The mean age of participants in this study was 61.99 years old. More than 70% of participants were rural residents. With regard to education level, illiterate participants accounted for 27.58% in rural and 9.33% in urban. Rural and urban participants living with others constitute the greatest proportion in the sample (78.04% in rural and 79.18% in urban). The rates of having chronic diseases in the sample were 43.67% and 46.67% for rural and urban participants, respectively. In addition, 16.42% of rural participants and 16.77% of urban participants had at least one outpatient visit in the past one month, respectively; 16.67% of rural participants and 17.91% of urban participants had at least one inpatient visit in the last year, respectively. Only 4.62% of rural participants and 2.76% of urban participants received paid family doctor services, respectively.

Table 3 shows the distribution of degree of dissatisfaction with local medical services: when dissatisfaction was treated as a continuous variable, the higher the score, and the more dissatisfied the participants. Overall, the mean score of dissatisfaction was 2.73 ± 1.08. Both urban and rural participants utilizing outpatients in the past month expressed higher dissatisfaction with local medical services than not utilized participants, while participants utilizing paid family doctor services showed less significantly (*p* < 0.05); male and participants having chronic diseases expressed more dissatisfaction (*p* < 0.001); in addition, the score of dissatisfaction increased with the decreasing of age and increasing of education level (*p* < 0.001). Table 4 also provides a comparison between dissatisfied and non-dissatisfied participants. Some characteristics of dissatisfied and non-dissatisfied groups for both rural and urban showed statistically significant differences. Participants were generally more dissatisfied with local medical services when they did not utilize paid family doctor services (rural: 15.63% vs. 10.14%, *p* < 0.001; urban: 19.29% vs. 9.92%, *p* = 0.007). Males appeared to be more dissatisfied with medical services than females (rural: 17.22% vs. 13.70%, *p* < 0.001; urban: 20.48% vs. 17.74%, *p* = 0.017). This is not surprising given that participants having chronic diseases have a higher degree of dissatisfaction (rural: 17.48% vs. 13.73%, *p* < 0.001; urban: 20.98% vs. 17.29%, *p* = 0.001). Moreover, urban participants utilizing more outpatient in the past one month and inpatient services in the last year tended to be more dissatisfied with local medical services (outpatient: 23.30% vs. 18.13%, *p* = 0.001; inpatient: 22.66% vs. 18.21%, *p* = 0.003), while there was no significant association among rural participants (*p* > 0.05).

### 3.2. Regression Results

Table 5 presents the results of multilevel mixed-effect models (model 1 for binary outcome of dissatisfaction and model 2 for continuous outcome of dissatisfaction). In rural, model 1 showed the participants’ utilization of paid family doctor services decreased the risk of occurring dissatisfaction (OR: 0.58, 95% CL: 0.45, 0.76); compared with males, the risk of occurring dissatisfaction decreased by 25% for females (OR: 0.75, 95% CL: 0.67, 0.83); the risk of occurring dissatisfaction decreased with the increasing of economic status, and the risk decreased by 16% for participants having middle economic status (OR: 0.84, 95% CL: 0.75, 0.94) and 27% for participants having high economic status (OR: 0.73, 95% CL: 0.63, 0.85), respectively; however, the risk of dissatisfaction was nearly 1.5 times for participants having chronic diseases compared with not having chronic diseases (OR: 1.33, 95% CL: 1.20, 1.46). Unlike significant reasons for dissatisfaction found in model 1, model 2 found more reasons and retained the same tendency. The utilization of outpatient in the past month, education of more than elementary school and having chronic diseases were positively associated with the increasing dissatisfaction score (*p* < 0.05), while utilization of paid family doctor services, aged more than 51 years old and higher economic status were negatively associated with the increasing dissatisfaction score (*p* < 0.05).

In urban, model 1 found that the utilization of outpatient services (OR: 1.41, 95% CL: 1.17, 1.72), inpatient services (OR: 1.21, 95% CL: 1.00, 1.47) and having chronic diseases (OR: 1.23, 95% CL: 1.06, 1.44) increased the risk of dissatisfaction significantly; meanwhile, being female (OR: 0.80, 95% CL: 0.68, 0.93) and utilizing the paid family doctor services (OR: 0.46, 95% CL: 0.25, 0.84) decreased the risk of dissatisfaction. Model 2 also indicates that the rate of dissatisfaction gradually increased with utilization of outpatient services, more than middle school education and having chronic diseases, while utilizing paid family doctor services, being female and aged more than 71 years were shown to be protective factors to dissatisfaction (*p* < 0.05).

## 4. Discussion

A quantitative exploration of the current status and causes of dissatisfaction with local medical services will not only enable medical institutions to identify problems and improve their services in a targeted manner but also enhance patients’ trust and reliance on local healthcare, thus further facilitating timely access to care and promoting health for all [33]. Firstly, this study revealed the level of dissatisfaction of middle-aged and elderly with local medical services. Then, it investigated the specific reasons for dissatisfaction from the individual’s perspective and identified areas for improvement. Overall, 16% of middle-aged and elderly in China were dissatisfied with the local medical services, 38% of them were satisfied with the local medical services, 16% of rural participants and 19% of urban participants were dissatisfied with the services they received, respectively. This dissatisfaction was higher than an investigation conducted 10 years before in China (13%) [10] but lower than those of studies conducted in Heilongjiang Province of China [9]; it was higher than those of studies conducted in the public and private wing of the health services in Ethiopia [13], in Dutch university medical centers [34] and in other developed countries [1] but lower than those of studies conducted in Tanzania [35], among psychiatric outpatients in Singapore [36] and diabetes mellitus patients in Pakistan [16]. This discrepancy among different studies in China may be due to the differences in the study population. In the current study, the study population was the middle-aged and elderly, which might have reduced their patient satisfaction as reported by other studies that as age increased patient satisfaction decreased [13,32]. Therefore, attention to improving the medical experience of the middle-aged and elderly is important in reducing dissatisfaction with the local medical services. The distinction in the level of satisfaction among different countries may be due to a gap in health systems [37]. More research is needed to verify the conclusion.

Previous studies showed that medical services (such as doctor and nurses services, medical equipment, waiting time) had significant impacts on dissatisfaction [15,38,39,40]. In the analysis, our results find that participants’ utilization of outpatient, inpatient and paid family doctor services was one reason for dissatisfaction. Utilization of outpatient and inpatient services would increase the risk of occurring dissatisfaction. It is not difficult to understand that the more patients may utilize outpatient and inpatient services, the more likely they are to be dissatisfied due to the current status of crowded medical services and the attitude of the tired doctors they visit [41]. However, utilization of the paid family doctor services would decrease the risk of dissatisfaction. This is because research has shown that family doctors act as a bridge between patients and doctors, taking full account of the patient’s feelings, promoting the patient’s best interests in health care and helping the patient understand the outcome of the doctor’s visit to the maximum extent possible. Ultimately, this promotes a good patient experience [42]. Therefore, the promotion of family doctor services can effectively reduce patients’ dissatisfaction with the local medical services.

With regard to the individual-level characteristics, our results show that dissatisfaction was more focused on males. Females were usually less dissatisfied, and this was in line with other studies indicating that being female was related to a lower rate of being dissatisfied [10,13,16]. The study also revealed that age was negatively associated with the increase of dissatisfaction. This finding is consistent with that of a study conducted in the public hospitals of Ethiopia: patients who were aged 38–47 years were more satisfied than those 48 years old and older groups [13]. This higher dissatisfaction may be related to the rising expectations of patients with increasing age; it may also be related to the fact that increasing knowledge and experience reduces satisfaction. In this study, the population that attended more than elementary school was positively associated with the increasing dissatisfaction score, and this finding is consistent with studies that revealed that as educational status increased, patient satisfaction decreased [13]. In addition, having chronic diseases were found to be positively associated with the increase of the dissatisfaction score. From this finding, it can be inferred that improving health status might improve patient satisfaction.

## 5. Limitations

Our understanding of dissatisfaction with local medical services is limited by some factors. Firstly, the study was not supported by qualitative methods. Secondly, the determinants of dissatisfaction were limited by the pre-specified questions in the survey, and there could be some potential unobserved confounding factors for which we did not control. For instance, distance to hospital and attitude of the doctor. Thirdly, this study is a correlation analysis, and the results should not be interpreted as causal. However, we believe that the results measured by this nationally representative sample will provide useful insights into improving the quality and efficiency of local medical services in China.

## 6. Conclusions

This pooled analysis of 14,263 rural and 4898 urban populations suggests that dissatisfaction with the local medical services was more focused on males, participants not utilizing the paid family doctor services and participants with chronic diseases. Therefore, promotion of family doctor services can effectively reduce patients’ dissatisfaction with the local medical services. In addition, more attention should be focused on males and participants with chronic diseases in order to decrease dissatisfaction with the local medical services. Studying the factors behind patients’ dissatisfaction with the local medical services is important for the provision of services as per patient needs; it helps hospital managers to improve the service experience and quality in a targeted manner based on the reasons. However, the findings of this study can only suggest an association between these reasons and patients’ dissatisfaction and cannot be interpreted as causal; more evidence based on a combination of quantitative studies (e.g., experimental study) and qualitative studies is needed to validate the causality in the next research steps.

## Figures and Tables

**Figure 1 ijerph-18-03931-f001:**
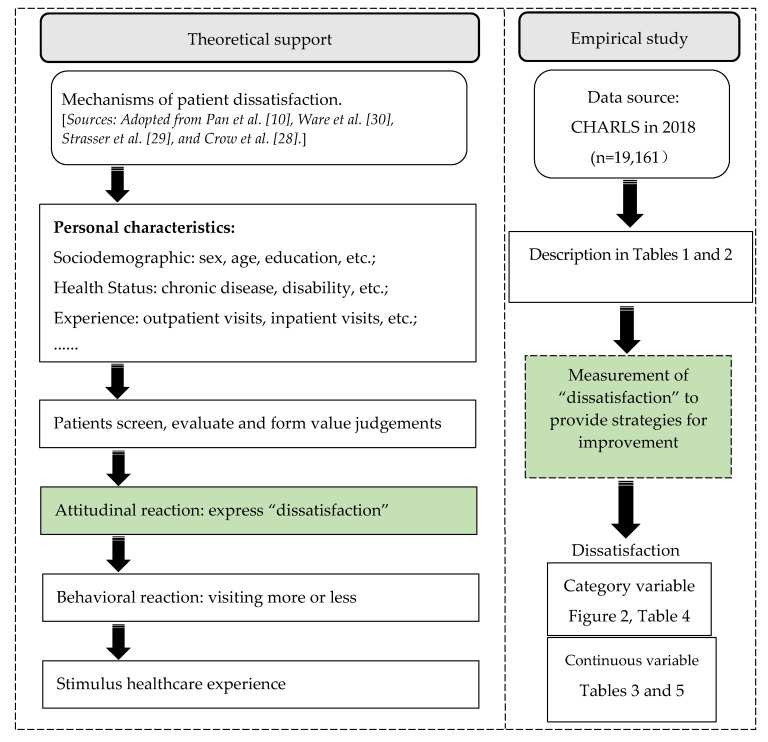
Flowchart of the study.

**Figure 2 ijerph-18-03931-f002:**
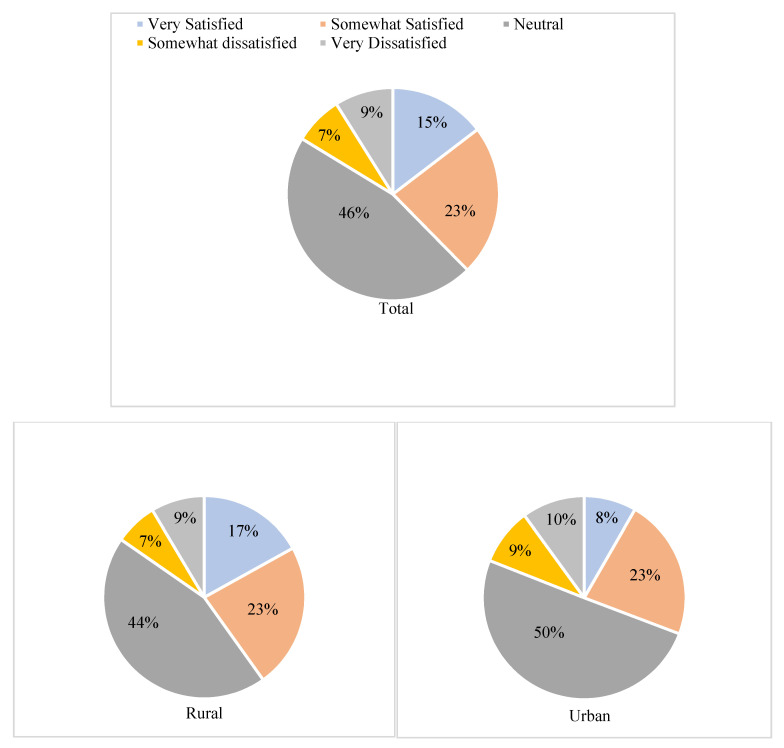
The distribution of participants’ attitude to local medical services.

**Table 1 ijerph-18-03931-t001:** Values assigned to the independent variables in multilevel modeling.

Variables	Description and Value
Dissatisfaction	Binary variable: 0 = Yes (somewhat dissatisfied and very dissatisfied); 1 = No (very satisfied, somewhat satisfied and neutral)
	Continuous variable, ranging from score 1 to 5
Type of residence	0 = Rural; 1 = Urban
Sex	0 = Male; 1 = Female
Age (years)	1 = 45–50; 2 = 51–60; 3 = 61–70; 4 = ≥71
Education	0 = Illiterate; 1 = ≤Elementary school; 2 = ≥Middle school
Living status	0 = Live with others; 1 = Live alone
Economic status	1 = Low; 2 = Middle; 3 = High
Chronic diseases	0 = No; 1 = Yes
Utilizing outpatients in the last month	0 = No; 1 = Yes
Utilizing inpatients in the past year	0 = No; 1 = Yes
Utilizing paid family doctor services	0 = No; 1 = Yes

**Table 2 ijerph-18-03931-t002:** Description of characteristics (*n* = 19,161).

Variables	Rural (*n* = 14,263)		Urban (*n* = 4898)		Total
	Frequency	Percent	Frequency	Percent	
Sex					
Male	6772	47.48	2326	47.49	9098
Female	7491	52.52	2572	52.51	10,063
Age (years)					
45–50	1749	12.26	679	13.86	2428
51–60	4596	32.22	1716	35.03	6312
61–70	4752	33.32	1513	30.89	6265
≥71	3166	22.20	990	20.22	4156
Education					
Illiterate	3934	27.58	457	9.33	4391
≤Elementary school	6694	46.93	1524	31.11	8218
≥Middle school	3635	25.49	2917	59.55	6552
Living status					
Live with others	11,131	78.04	3878	79.18	15,009
Live alone	3132	21.96	1020	20.82	4152
Economic status					
Low	4143	29.05	444	9.06	4587
Middle	7536	52.84	2089	42.65	9625
High	2583	18.11	2365	48.29	4948
Chronic diseases					
No	8034	56.33	2612	53.33	10,646
Yes	6229	43.67	2286	46.67	8515
Utilizing outpatients					
No	11,920	83.58	4075	83.23	15,995
Yes	2342	16.42	821	16.77	3163
Utilizing inpatients					
No	11,885	83.33	4019	82.09	15,904
Yes	2377	16.67	877	17.91	3254
Utilizing paid family doctor services					
No	13,603	95.38	4760	97.24	18,363
Yes	659	4.62	135	2.76	794

**Table 3 ijerph-18-03931-t003:** Distribution of degree of dissatisfaction with local medical services.

Variables	Rural (*n* = 14,263)			Urban (*n* = 4898)		
	Mean ± SD	t/F	*p*	Mean ± SD	t/F	*p*
Utilizing outpatients						
No	2.65 ± 1.10	−3.42	<0.001	2.88 ± 1.00	−2.90	0.002
Yes	2.74 ± 1.08			2.99 ± 1.08		
Utilizing inpatients						
No	2.67 ± 1.09	0.20	0.420	2.89 ± 1.01	−1.77	0.039
Yes	2.67 ± 1.13			2.95 ± 1.07		
Utilizing paid family doctor services						
No	2.68 ± 1.10	6.79	<0.001	2.91 ± 1.02	4.52	<0.001
Yes	2.38 ± 1.04			2.50 ± 1.00		
Sex						
Male	2.74 ± 1.11	7.10	<0.001	2.97 ± 1.03	4.14	<0.001
Female	2.60 ± 1.09			2.84 ± 1.01		
Age (years)						
45–50	2.80 ± 0.99	31.74	<0.001	2.95 ± 0.98	3.31	0.019
51–60	2.75 ± 1.07			2.88 ± 0.98		
61–70	2.62 ± 1.13			2.95 ± 1.07		
≥71	2.55 ± 1.13			2.83 ± 1.04		
Education						
Illiterate	2.49 ± 1.16	78.57	<0.001	2.70 ± 1.16	14.70	<0.001
≤Elementary school	2.70 ± 1.08			2.85 ± 1.06		
≥Middle school	2.79 ± 1.04			2.96 ± 0.97		
Living status						
Live with others	2.68 ± 1.09	1.62	0.053	2.91 ± 1.00	1.77	0.039
Live alone	2.64 ± 1.13			2.85 ± 1.10		
Economic status						
Low	2.68 ± 1.15	1.08	0.341	2.77 ± 1.18	3.98	0.019
Middle	2.66 ± 1.11			2.93 ± 1.04		
High	2.68 ± 1.00			2.90 ± 0.97		
Chronic diseases						
No	2.62 ± 1.07	−5.73	<0.001	2.84 ± 1.01	−3.91	<0.001
Yes	2.73 ± 1.13			2.96 ± 1.02		

**Table 4 ijerph-18-03931-t004:** Distribution of rate of dissatisfaction with local medical services.

Variables	Rural (*n* = 14,263)			Urban (*n* = 4898)		
	No	Yes	*p*	No	Yes	*p*
Utilizing outpatients						
No	9711 (84.83)	1736 (15.17)	0.130	3165 (81.87)	701 (18.13)	0.001
Yes	1925 (83.59)	378 (16.41)		622 (76.70)	189 (23.30)	
Utilizing inpatients						
No	9700 (84.83)	1734 (15.17)	0.131	3118 (81.79)	694 (18.21)	0.003
Yes	1936 (83.59)	380 (16.41)		669 (77.34)	196 (22.66)	
Utilizing paid family doctor services						
No	11,051 (84.37)	2048 (15.63)	<0.001	3669 (80.71)	877 (19.29)	0.007
Yes	585 (89.86)	66 (10.14)		118 (90.08)	13 (9.92)	
Sex						
Male	5413 (82.78)	1126 (17.22)	<0.001	1751 (79.52)	451 (20.48)	0.017
Female	6223 (86.30)	988 (13.70)		2036 (82.26)	439 (17.74)	
Age (years)						
45–50	1461 (85.64)	245 (14.36)	0.224	519 (81.09)	121 (18.91)	0.007
51–60	3739 (84.02)	711 (15.98)		1370 (83.03)	280 (16.97)	
61–70	3882 (84.32)	722 (15.68)		1135 (78.17)	317 (21.83)	
≥71	2554 (85.42)	436 (14.58)		763 (81.60)	172 (18.40)	
Education						
Illiterate	3231 (86.30)	513 (13.70)	0.001	351 (82.78)	73 (17.22)	0.568
≤Elementary school	5477 (84.50)	1005 (15.50)		1180 (81.10)	275 (18.90)	
≥Middle school	2928 (83.09)	596 (16.91)		2256 (80.63)	542 (19.37)	
Living status						
Live with others	9143 (84.62)	1662 (15.38)	0.964	3038 (81.36)	696 (18.64)	0.177
Live alone	2493 (84.65)	452 (15.35)		749 (79.43)	194 (20.57)	
Economic status						
Low	3305 (83.06)	674 (16.91)	0.001	337 (81.20)	78 (18.80)	0.391
Middle	6168 (84.85)	1101 (15.15)		1591 (80.07)	396 (19.93)	
High	2162 (86.45)	339 (13.55)		1859 (81.71)	416 (18.29)	
Chronic diseases						
No	6655 (86.27)	1059 (13.73)	<0.001	2047 (82.71)	428 (17.29)	0.001
Yes	4981 (82.52)	1055 (17.48)		1740 (79.02)	462 (20.98)	

**Table 5 ijerph-18-03931-t005:** Determinants of dissatisfaction with local medical services by multilevel mixed-effect models.

Variables	Rural						Urban					
	Model 1			Model 2			Model 1			Model 2		
	OR	95% CL	*p*	*β*	95% CL	*p*	OR	95% CL	*p*	*β*	95% CL	*p*
Utilizing outpatients	1.06	0.93, 1.21	0.357	0.07	0.02, 0.12	0.008	1.41	1.17, 1.72	<0.001	0.12	0.04, 0.20	0.002
Utilizing inpatients	1.02	0.90, 1.16	0.775	−0.003	−0.05, 0.05	0.910	1.21	1.00, 1.47	0.056	0.04	−0.04, 0.11	0.323
Utilizing paid family doctor services												
	0.58	0.45, 0.76	<0.001	−0.31	−0.39, −0.22	<0.001	0.46	0.25, 0.84	0.011	−0.39	−0.57, −0.22	<0.001
Female	0.75	0.67, 0.83	<0.001	−0.11	−0.15, −0.07	<0.001	0.80	0.68, 0.93	0.004	−0.11	−0.17, −0.06	<0.001
Age (51–60 years)	1.07	0.91, 1.26	0.417	−0.07	−0.13, −0.01	0.017	0.85	0.67, 1.10	0.225	−0.07	−0.16, 0.02	0.120
Age (61–70 years)	1.05	0.89, 1.24	0.576	−0.18	−0.24, −0.12	<0.001	1.14	0.87, 1.46	0.311	−0.03	−0.12, 0.07	0.602
Age (≥71 years)	0.95	0.79, 1.14	0.580	−0.24	−0.31, −0.17	<0.001	0.89	0.67, 1.18	0.410	−0.11	−0.22, −0.01	0.035
≤Elementary school	1.07	0.94, 1.22	0.282	0.13	0.08, 0.18	<0.001	1.08	0.79, 1.46	0.625	0.10	−0.01, 0.22	0.064
≥Middle school	1.18	1.01, 1.38	0.036	0.19	0.13, 0.25	<0.001	1.17	0.86, 1.59	0.304	0.20	0.09, 0.32	<0.001
Live alone	1.04	0.92, 1.17	0.526	0.02	−0.03, 0.06	0.443	1.19	0.98, 1.44	0.083	−0.03	−0.10, 0.05	0.492
Middle economic status	0.84	0.75, 0.94	0.003	−0.06	−0.10, −0.01	0.008	0.99	0.75, 1.33	0.927	0.08	−0.02, 0.19	0.120
High economic status	0.73	0.63, 0.85	<0.001	−0.09	−0.14, −0.03	0.004	0.86	0.64, 1.15	0.298	0.01	−0.10, 0.12	0.878
Chronic diseases	1.33	1.20, 1.46	<0.001	0.12	0.09, 0.16	<0.001	1.23	1.06, 1.44	0.009	0.12	0.06, 0.18	<0.001

Model 1: binary outcome of dissatisfaction; Model 2: continuous outcome of dissatisfaction; OR: odds ratios; CL: confidence limits; β: coefficient.

## Data Availability

The datasets and questionnaire are available at http://charls.pku.edu.cn/ (accessed on 23 September 2020).

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
