# Peer review of "Dissatisfaction with Local Medical Services for Middle-Aged and Elderly in China: What Is Relevant?"

_ijerph, 2021, doi:10.3390/ijerph18083931_

Round 1

Reviewer 1 Report

            The paper Dissatisfaction with local medical services for middle-aged and elderly in China: What is relevant? provides an analysis of the main determinants of dissatisfaction with respect to the medical services offered to middle-aged and elderly patients from the urban and rural areas of 150 counties and 450 villages across 28 provinces in China. By using data collected from a survey conducted on a sample of 19,161 participants, whose perception and opinions were retrieved from the China Health and Retirement Longitudinal Study published in 2018, the authors measured the risk of occurring dissatisfaction, by means of binary outcome method, and the degree of dissatisfaction, by means of another multilevel model, i.e. the continuous outcome method. The research article aims at being published in a well reputed economic journal, International Journal of Environmental Research and Public Health.

            The article is well structured, it fulfils to a certain extent the requirements made by the editorial board, it is interesting and of a high relevance in the context of a continuous need for healthcare services, especially in the pandemic context we are going through. It seems that the relevance of the paper is given also by the actual case study made on China, a country which is not only the most populated on Earth (17.5% of the worldwide population), but which is also experiencing a gradual increase in ageing and for which the middle-aged population and especially the elderly citizens might encounter serious problems with respect to maintaining their health in good parameters or improving their health status if necessary.

            Consistency

            The approach presented in the paper seems to be an adequate one. The general background is well presented and the references mentioned in the introduction part and in the following sections seem to cover a worldwide perspective concerning the analysis of the determinants for the dissatisfaction felt among many patients treated by family doctors all over China. Still, I do have a comment on the distribution of the interviewed people considered in the analysis between rural and urban areas. Out of the total number of 19,161 participants, only 4,898 (25.56%) have their residence in an urban area. I think the number of people coming from urban and rural areas, respectively, should have been more balanced in order to obtain much more relevant results for your study. I do not have the information, but if about ¾ of the total population lives in the countryside, then your sample might be ok. I have a similar remark regarding the healthcare utilization characteristics, especially the utilizing paid family doctor services indicator which seems to be disproportionate. Nevertheless, the distribution of the sample on the other independent variables describing the socio-demographic characteristics considered for the analysis seems to be balanced enough.

In the Introduction part, I would suggest to even more clearly state what makes the paper original, which are the elements that bring about the novelty of the research. There are several relevant remarks introduced in the text like “there is limited evidence from both urban and rural China focusing on the patient dissatisfaction and to what degree patient characteristics explains dissatisfaction among middle-aged and elderly” or “research on dissatisfaction among middle-aged and elderly would be an important reference for targeted efforts to improve the health care experience…” etc., but the authors should also mention precisely in the text that all these above mentioned arguments make their research paper an original piece of work. I would also suggest that the authors briefly present the content of the following sections of the paper in the end of this section (introduction).

After analysing the specific literature, the authors applied a complex scheme of analysis that brought about a consistent study. The methodology and the materials used are pretty well described in the article, as well as the results and discussion, which are accurately and clearly presented. The authors also emphasized the actual economic, social and political-administrative background that explains the results pinpointed in the last sections of the paper. There is also a strong connection between the main results of the paper and the results reached in other papers concerning the same topic. In addition, I appreciate the identification of the limitations of the study and the fact that they were properly mentioned.

However, the conclusion is a little bit too short. One or two more paragraphs should be added. For instance, some future research directions or improvements should be properly highlighted in the conclusion section.

              Technical issues

            According to the International Journal of Environmental Research and Public Health (IJERPH) instructions for authors, the abstract should be of about 200 words maximum. The paper has 257 words, which I consider to be a little bit too much. I would suggest rephrasing while taking care of not overpassing the limit required and be more consistent. For instance, I would not mention in the abstract the actual OR and CL values, which, by the way, what do they mean? The abstract is not meant for giving too many details of the actual analysis. For a simple reader who wants to get the main ideas of your paper from the abstract, its length is not that attractive, while the abbreviations create confusion at this point. No abbreviations should be used in the abstract, especially since they are quite specific to your analysis and not that much used on a large scale. The first keyword (“Dissatisfaction”) should be written with a lowercase. At the same time, I would not keep just the words middle-aged or elderly, I would add “person” or “patients” after them. It is just a suggestion. Perhaps you could also consider adding some other keywords, such as multilevel analysis, binary outcome, continuous outcome etc.

            When you refer to a country, avoid using the particle “the” before the name of the country (see for instance “A study in the Switzerland” (page 2), “In the Pakistan” (page 2)). In addition, perhaps it would be better to avoid also using the actual name of the authors (Aziz et al., Jalil, Ruggieri etc.) included in the literature review and keep only the consecutive numbers placed between square brackets before punctuation.

            Please check whether “n=19,441” which appears between brackets in the title of Table 2 on page 7 is correct.

            The text on page 10, after Table 4, seems to be wrongly positioned on the page (too much indent) so that not all the text is readable.

            The references at the end of the paper do not respect entirely the requirements made by the editor. For instance:

   - between the last name of the author and the initial of the first name there should be a comma;

   - after each initial of the first name a full stop “.” should be placed;

   - instead of “:” after the names of all authors, there should be a “.”;

   - use italic font for the volume of the journal of the cited article;

   - write “p.” or “pp.” for introducing the pages of the cited article into the journal;

   - do not use “:” before introducing the pages of an article or chapter in a book, only comma “,” or “;” if it is a book.

            None of the 35 reference notes contains information about DOI, as it is required (where available).

Table 1 should be placed after the paragraph it was first mentioned in 2.2. subsection, where the variables are presented and described, not in 2.3. subsection which refers to the data analysis. Before the title of the second figure there are some blank lines left which should be eliminated. The text describing the content of Table 3 and Table 4 should be split by introducing Table 3 for instance so that the two of them should not come immediately one after the other on page 9 and 10. The tables appear too late as compared to their first mentioning in the text. This happens also for Table 5, which I consider it should be placed somewhere in the actual text explaining the determinants of dissatisfaction and not in the end of subsection 3.2.

             Finally, based on the previous conclusions resulted after reading the paper, I answer some important evaluation questions:

Did the researcher carry out his/her own research project, used his/her own results?

Yes, the paper seems to be an original piece of work, but the author/s could better pinpoint this issue.

Is the subject of the paper relevant and interesting for the readers? The motivation for the paper should include a clearly specified research question and a statement as to why this question is interesting.

The subject is relevant and the research questions are clearly stated.

Is the overview of relevant literature of the topic well summarised, are all important research background results referred to?

The literature review is directly related to the research and well summarised.

Is the research paper clear, logic and well structured?

The paper seems to be well structured.

Does the author use the English of the specific field correctly?

The paper is written in a clear and correctly specific field English and, generally speaking, it is grammatically correct in my view. It needs only some minor checking, for instance when mentioning the names of the countries.

Does the research method used by author fit the subject? Are there new, interesting aspects of the research method used in the study?

The methodology is properly explained in the paper: methods, study area and data source.

Does the author arrive at original or new points as to the conclusions of the paper?

The results are interesting and relevant, but the general conclusions could be improved. Some similarities and differences between this particular case in China and other international cases are also underlined.

Do you, in summary, propose the paper for publication as it is/ publication with changes suggested (please list changes in detail)/ or outright rejection.

Due to the relevance of the paper, to the consistent methodology and analysis and to the identification of only minor issues, I suggest the acceptance of the paper for being published.

Author Response

Response to reviewer 

Consistency

The approach presented in the paper seems to be an adequate one. The general background is well presented and the references mentioned in the introduction part and in the following sections seem to cover a worldwide perspective concerning the analysis of the determinants for the dissatisfaction felt among many patients treated by family doctors all over China.

(1) I do have a comment on the distribution of the interviewed people considered in the analysis between rural and urban areas. Out of the total number of 19,161 participants, only 4,898 (25.56%) have their residence in an urban area. I think the number of people coming from urban and rural areas, respectively, should have been more balanced in order to obtain much more relevant results for your study. I do not have the information, but if about ¾ of the total population lives in the countryside, then your sample might be ok. I have a similar remark regarding the healthcare utilization characteristics, especially the utilizing paid family doctor services indicator which seems to be disproportionate. Nevertheless, the distribution of the sample on the other independent variables describing the socio-demographic characteristics considered for the analysis seems to be balanced enough.

Response 1: Thanks for your comment. Firstly, the majority of people in China are rural type of residence and therefore there are more rural than urban people in this study. Secondly, the disproportionate data on the utilizing paid family doctor services indicator may be due to the fact that the data was collected (in 2018) shortly after the implementation of the paid family doctor services policy (in June 06, 2016) and therefore the findings need to be confirmed by further research.

(2) In the Introduction part, I would suggest to even more clearly state what makes the paper original, which are the elements that bring about the novelty of the research. There are several relevant remarks introduced in the text like “there is limited evidence from both urban and rural China focusing on the patient dissatisfaction and to what degree patient characteristics explains dissatisfaction among middle-aged and elderly” or “research on dissatisfaction among middle-aged and elderly would be an important reference for targeted efforts to improve the health care experience…” etc., but the authors should also mention precisely in the text that all these above mentioned arguments make their research paper an original piece of work. I would also suggest that the authors briefly present the content of the following sections of the paper in the end of this section (introduction).

Response 2: We have added a description on the innovation and originality of the study in response to your comments (lines 105-106), and we also briefly presented the content of the following sections of the paper in the end of the introduction section (lines 117-120).

(3) The conclusion is a little bit too short. One or two more paragraphs should be added. For instance, some future research directions or improvements should be properly highlighted in the conclusion section.

Response 3: Thanks for your comment. We have expended the conclusion section by including suggestions for the future works (lines 391-394).

“However, the findings of this study can only suggest an association between these reasons and patients’ dissatisfaction and cannot be interpreted as causal, more evidences based on a combination of quantitative studies (e.g. experimental study) and qualitative studies are needed to validate the causals in the next research steps.”

Technical issues

(4) According to the International Journal of Environmental Research and Public Health (IJERPH) instructions for authors, the abstract should be of about 200 words maximum. The paper has 257 words, which I consider to be a little bit too much. I would suggest rephrasing while taking care of not overpassing the limit required and be more consistent. For instance, I would not mention in the abstract the actual OR and CL values, which, by the way, what do they mean? The abstract is not meant for giving too many details of the actual analysis. For a simple reader who wants to get the main ideas of your paper from the abstract, its length is not that attractive, while the abbreviations create confusion at this point. No abbreviations should be used in the abstract, especially since they are quite specific to your analysis and not that much used on a large scale. The first keyword (“Dissatisfaction”) should be written with a lowercase. At the same time, I would not keep just the words middle-aged or elderly, I would add “person” or “patients” after them. It is just a suggestion. Perhaps you could also consider adding some other keywords, such as multilevel analysis, binary outcome, continuous outcome etc.

Response 4: We have revised the abstract according to your suggestions (lines 29-32, 37-38).

(5) When you refer to a country, avoid using the particle “the” before the name of the country (see for instance “A study in the Switzerland” (page 2), “In the Pakistan” (page 2)). In addition, perhaps it would be better to avoid also using the actual name of the authors (Aziz et al., Jalil, Ruggieri etc.) included in the literature review and keep only the consecutive numbers placed between square brackets before punctuation.

Response 5: We have revised these contents in the manuscript (lines 58-83).

(6) Please check whether “n=19,441” which appears between brackets in the title of Table 2 on page 7 is correct.

Response 6: We have corrected this information in Table 2.

(7) The text on page 10, after Table 4, seems to be wrongly positioned on the page (too much indent) so that not all the text is readable.

Response 7: Thanks for your comment. We have reformatted the content to ensure that they are all clearly readable (page 11).

(8) The references at the end of the paper do not respect entirely the requirements made by the editor. For instance:

   - between the last name of the author and the initial of the first name there should be a comma;

   - after each initial of the first name a full stop “.” should be placed;

   - instead of “:” after the names of all authors, there should be a “.”;

   - use italic font for the volume of the journal of the cited article;

   - write “p.” or “pp.” for introducing the pages of the cited article into the journal;

   - do not use “:” before introducing the pages of an article or chapter in a book, only comma “,” or “;” if it is a book.

Response 8: We have revised the references style according to the requirements made by the editor (lines 417-506).

 (9) None of the 35 reference notes contains information about DOI, as it is required (where available).

Response 9: Thanks for your comment. We have revised the references style according to the instruction to authors (lines 417-506).

(10) Table 1 should be placed after the paragraph it was first mentioned in 2.2. subsection, where the variables are presented and described, not in 2.3. subsection which refers to the data analysis. Before the title of the second figure there are some blank lines left which should be eliminated. The text describing the content of Table 3 and Table 4 should be split by introducing Table 3 for instance so that the two of them should not come immediately one after the other on page 9 and 10. The tables appear too late as compared to their first mentioning in the text. This happens also for Table 5, which I consider it should be placed somewhere in the actual text explaining the determinants of dissatisfaction and not in the end of subsection 3.2.

Response 10: Thanks for your comment. We have adjusted the position of the table to your suggestion.

Reviewer 2 Report

This paper presents a discussion on services that can reduce middle-aged and elderly’ dissatisfaction with local medical services. The manuscript is well written and organized and should be accepted for publication in this journal. However, I suggest a careful review of the text to avoid grammar, spelling, and punctuation errors as well as typos. For acceptance, it is necessary to include the approval of a research ethics committee as well. The introduction could be more strong and more referenced. I suggest include some important surveys of the current state of the art. Improve the Figure 2 presentation. Include suggestions for future works.

Author Response

Response to reviewer 

(1) This paper presents a discussion on services that can reduce middle-aged and elderly’ dissatisfaction with local medical services. The manuscript is well written and organized and should be accepted for publication in this journal. However, I suggest a careful review of the text to avoid grammar, spelling, and punctuation errors as well as typos. For acceptance, it is necessary to include the approval of a research ethics committee as well.

Response 1: Thanks for your comments. Our revised manuscript has been reviewed by one of our co-author Duolao Wang to avoid grammar, spelling, and punctuation errors, who is fluent with English.

The approval of a research ethics committee has been added in the manuscript (lines: 405-406).

(2) The introduction could be more strong and more referenced. I suggest include some important surveys of the current state of the art.

Response 2: We have revised the introduction and added some references based on your suggestions (lines 58-84).

“Based on a literatures search, we found that most dissatisfaction surveys are based on quantitative cross-sectional studies [2, 5, 8, 9], cohort studies [10], or qualitative interviews [11]. Several studies have investigated that patient satisfaction is shaped by some determinants including personal characteristics, preference, expectation, the quality of care received in other countries based on cross-sectional surveys [5, 12-14]. In addition, A cross-sectional study in Switzerland found that the most satisfied patients were the most likely to participate in a post-hospitalization satisfaction survey [8]. In Pakistan, a low level of patient satisfaction was found to be correlated with the medical services [15]; scholars found that the developing of the interpersonal and clinical skills of doctors could improve patient satisfaction [16]. Cross-sectional study in Italian university hospitals also found that lower patient satisfaction were associated with a higher patients leaving hospital against medical advice rates [2]. Studies emerged from China recently are only limited to selected groups, tertiary public hospital, certain types of treatments or health conditions [17]. For example, in a sample of 300 inpatients in one tertiary public hospital found that the discharge and admission process does not yet meet patient expectations [18]. To investigate provider related factors, a cross-sectional study found that higher level of trust, lower levels of hospital medical expenditure and good staff attitude were key predictors of patient satisfaction [9]. Some scholars studied the impact of both patient- and provider-related characteristics on patient satisfaction in urban China by a cohort study [10]. ”

(3) Improve the Figure 2 presentation.

Response 3: We have revised Figure 2.

(4) Include suggestions for future works.

Response 4: We have added suggestions in the conclusion section according to your suggestion (lines 391-394).

“However, the findings of this study can only suggest an association between these reasons and patient’ dissatisfaction and cannot be interpreted as causal, more evidences based on a combination of quantitative studies (e.g. experimental study) and qualitative studies are needed to validate the causals in the next research steps.”

Reviewer 3 Report

Thank you for the opportunity to review this paper. The topic is very interesting and topical all over the world and the paper provides an aid for thought.

My notes follow:
In the introduction the choice of sample is not explained, why do the authors investigate a middle aged and elderly people sample?
In the introduction it is difficult to understand the necessity of the study.

The conclusion lacks practical proposals for the improvement of services and for further research.

Author Response

Response to reviewer 

 (1) In the introduction the choice of sample is not explained, why do the authors investigate a middle aged and elderly people sample?

Response 1: Thanks for your question. We have provided the following description of reasons for choosing the middle-aged and elderly population in the revised manuscript (Introduction section, lines 89-105).

“With the rapid development of Chinese economy, people aged 60 years and over accounted for 13.26% of the total according to the result of the six national census, up 2.93 percentage points from the fifth national census, and 8.87% of people aged 65 and over, up 1.91 percentage points from the fifth national census [18], the ageing process is beginning to accelerate gradually. Studies have found that ageing largely alters the functional capacity of the human body [19, 20] and is strongly associated with the incidence of common diseases, chronic illnesses and cancer [21-23]. With the changing age structure of the world’s population and the gradual increase in ageing in China, the middle-aged and elderly population will make up the majority of the groups utilizing hospital healthcare services in the future, research on dissatisfaction among them would be an important reference for targeted efforts to improve the health care experience of the target population, promote a good health care environment and reduce social risks.”

(2) In the introduction it is difficult to understand the necessity of the study.

Response 2: Thanks for your comment. We have reorganized the necessity of the study in the introduction section (lines 107-117).

“In this study, we explored patients’ dissatisfaction with local health services among middle-aged and elderly people by adopting a nationally representative cross-sectional survey data in China. The findings will uncover the level of patients’ dissatisfaction from the largest and most populated developing countries, investigate the relationship between patients’ dissatisfactions and their characteristics (such as socio-demographic characteristics and health services utilization) in rural and urban area and reveal the relative significant reasons to dissatisfaction. The ultimate aim is to help managers in hospitals and policy-makers to take target measures for improving patient satisfaction.”

(3) The conclusion lacks practical proposals for the improvement of services and for further research.

Response 3: Thank you for your suggestions. The improvement of services has been descripted in the conclusion section (lines 384-388). We have added some suggestion for further research (lines 391-394).

“However, the findings of this study can only suggest an association between these reasons and patient’ dissatisfaction and cannot be interpreted as causal, more evidences based on a combination of quantitative studies (e.g. experimental study) and qualitative studies are needed to validate the causals in the next research steps.”

Round 2

Reviewer 3 Report

I would like to thank the authors for following the instructions. I have no further indications